# OpenReview forum: "Self-rationalization improves LLM as a fine-grained judge"
_ICLR.cc/2025/Conference — Submitted to ICLR 2025_

### Official Review · Reviewer_DnQV · 2024-10-30

**Soundness:** 3
**Presentation:** 3
**Contribution:** 2
**Rating:** 5
**Confidence:** 4

**Summary:**

This paper trains an LLM-as-judge model to generate both point-wise  and pairwise assessments. The model is fine-tuned on a training dataset that has a context, response, the rating of the response and the rationale (chain of thought reasoning) used to arrive at the rating. Upon training the model, a subset of the training data is sampled and for each datapoint, multiple generations of rationale and score are obtained. These generations are used to create the preference data for DPO. This iterative DPO is performed twice.

A preference pair for the DPO is created  using the following techniques: (1) Amongst all the generations, the generation whose score matches the ground truth is taken as the chosen judgement and any of the incorrect one is taken as the rejected one. (2) The generations whose score is the mode are considered as the chosen ones while the others are considered as the rejected ones.  (3) A meta-judge is asked to rate either the rationales or the scores. Its rating is used to create the preference pairs where for each pair, the chosen judgement is rated higher than the rejected judgement. Several preference pairs are created for each datapoint.

**Strengths:**

They are able to take custom criteria and generating rating in accordance to it. Their evaluation datasets for point-wise comparison are very extensive and compare over multiple domains. They also empirically demonstrate that the iterative DPO helps in generating good quality rationales when compared to SFT.

**Weaknesses:**

1. This paper's ideas are heavily borrowed  from the self rewarding LLMs. That paper also uses rationale and scores to train a base LLM-as-a-judge model. After that it creates preference data based on the model's generations and performs iterative DPO. The main difference is that the self rewarding LLMs model is designed to only generate pairwise judgement and is trained in a multitask fashion to perform both LLM-as-a-judge and to follow instructions. While your paper uses the seed training data to create the preference data, the self rewarding paper asks the model itself to generate synthetic data. Though the initial model used in self-rewarding paper is a llama2 70B model, the algorithm can be applied to llama 3.1 8B model as well. Despite these variations, the basic methodologies remain the same in both the works. Thus, the novelty of this paper is fairly limited. Can you please describe the contributions of this paper when compared to the self rewarding LLMs paper.

2.  Given that the model was trained on both pairwise and point-wise comparison tasks, please include evaluations for pairwise comparison task on datasets such as RewardBench, PreferenceBench, InstruSum [1], HHH [2]  etc. Include https://huggingface.co/NCSOFT/Llama-3-OffsetBias-8B and a llama 8 version of self rewarding LLMs also as a baseline in addition to Skywork-Critic-Llama-3.1-8B.

[1] Benchmarking generation and evaluation capabilities of large language models for instruction controllable summarization, Liu et al.
[2] A general language assistant as a laboratory for alignment, Askell et al.

**Questions:**

1. Why did you perform model merging? What were the challenges with multi-task training the initial model to perform both the tasks?
2. What is the time taken to train a self-rationalizing model vs the other baselines?

---

> ### Author Response · Authors · 2024-11-21
> **Rebuttal by authors to Reviewer DnQV**
>
> We thank the reviewers for their detailed questions and comments We provide our answers point-wise and seek guidance/advice on making suggested changes in the revised version.  We have uploaded a version where new changes are marked with blue text color.
>
> > Comparison to self-rewarding LLMs
>
> Following are the differences between Self-rewarding LMs and our propose method SRE:
> - As mentioned in Table 1, Self-rewarding LMs requires additional training data for Instruction Fine-Tuning as well as Evaluation Fine-tuning, thus being much more data costly than SRE.
> - Self-rewarding LMs do not have the ability to evaluate on the basis of a custom as well as a fine-grained likert-scale criteria
> - As mentioned by the reviewer, they have not demonstrated improvements through their recipe on a small parameter model
>
> >Pairwise evaluations
>
> We sincerely thank the reviewer for their valuable suggestion to include pairwise benchmarks in the paper. In response, we have incorporated the corresponding results in Table 3 of the revised manuscript. These results demonstrate that J_SRE exhibits robust performance across both pairwise and pointwise benchmark evaluations, along with the added benefits of being a small sized LM(<10B) and having the ability to judge based on a custom scoring criteria.
>
> Regarding other baseline models, we have added Prometheus2-7B[1] for comparison as it is the closest model in terms of ability( it can also perform pairwise as well as pointwise assessments), gives rationale and is of similar size. As seen in Table 3, we outperform Prometheus2-7B as well as J_SFT and LLama-3.1-8B-Instruct.
> As Skywork-Critic-Llama-3.1-8B, Llama-3-OffsetBias-8B and Self-rewarding LMs do not have the features that we have( see Table 1 for more details), do you recommend adding them as baselines? We would be happy to clarify more if needed!
> | Model                               | Reward-Bench         |            |            |            | Total Score | Safe-RLHF       | Preference Bench |
> |-------------------------------------|----------------------|------------|------------|------------|-------------|-----------------|------------------|
> |                                     | Chat                 | Chat-hard  | Safety     | Reasoning  |             | Human Pearson   | GPT4 Pearson     |
> | LLama-3.1-8B-Instruct (seed model)  | 0.80                 | 0.49       | 0.64       | 0.68       | 0.65        | 0.51            | 0.63             |
> | SFT J_SFT                           | 0.88                 | 0.45       | 0.84       | 0.75       | 0.74        | 0.70            | 0.93             |
> | Prometheus-2 7B                     | 0.85                 | 0.49       | 0.77       | 0.76       | 0.72        | -               | 0.92             |
> | SRE J_SRE (margin ≥ 0)              | 0.89                 | 0.51       | 0.85       | 0.80       | 0.78        | 0.68            | 0.95             |
> | SRE J_SRE (margin ≥ 1)              | **0.92**             | **0.50**   | **0.84**   | **0.87**   | **0.83**    | **0.71**        | **0.96**         |
>
>
>
>
> > Weight merging vs Dataset mixtures
>
> Prometheus 2[1] have demonstrated experimentally the effect of single-format training, joint-training(i.e. Merging the datasets) and weight merging. Similar to their results we observed that linear merging the pairwise and the pointwise models improved the task performance of both tasks.
>
> >Training time
>
> Thanks for the feedback! As other baseline models( for example iterative baselines mentioned in Table 1) have not mentioned their training time as well as computation resources, we unfortunately cannot compare the training time with the baselines. But we can predict that the training time of the proposed recipe is much less than other iterative baselines as SRE required just 2 iterations to reach peak performance whereas other baseline models required more iterations as shown in table below:
> | Method                     | Number of Iterations |
> |----------------------------|----------------------|
> | SRE                        | 2                   |
> | Self-taught Evaluators  | 5                   |
> | Self-rewarding LMs     | 3                   |
> | Meta-rewarding LMs         | 4                   |
> | IRPO                       | 4                   |
>
> [1] Prometheus 2: An Open Source Language Model Specialized in Evaluating Other Language Models Kim et. al.

---

> > ### Comment · Reviewer_DnQV · 2024-11-26
> > **Response to rebuttal**
> >
> > I thank the authors for their rebuttal. The evaluations on pairwise comparison tasks demonstrate that the model is outperforming the other baselines
> > Reg the difference between Self-Rewarding LLMs and the current work : The current work also requires data to train 2 models on point-wise comparison and pairwise comparison respectively. I also pointed out the differences highlighted by the authors in my original review. While it is true that the Self-Rewarding LLMs used a much larger model and don't support a fine-grained custom criteria for point-wise evaluation, that is still not sufficient to differentiate the current work from Self-Rewarding LLMs. The core training paradigm is heavily borrowed from Self-Rewarding LLMs.  I updated my score

---

### Official Review · Reviewer_Dq8w · 2024-11-03

**Soundness:** 2
**Presentation:** 3
**Contribution:** 2
**Rating:** 3
**Confidence:** 5

**Summary:**

This work proposes Self-Rationalization for an iterative training scheme to achieve self-improving language models. Specifically, the authors show that training instances for the LLM-as-a-Judge task for self-evaluation can be obtained by oversampling and post-filtering of rationale trajectories after SFT even when no external preference dataset exists. In particular, they show that approaches such as majority voting, which is widely utilized in existing Chain-of-Thought research, can be used as a post-filtering approach to construct the LLM-as-a-Judge dataset, and also score margin-based filtering is effective. This work further demonstrated that preference dataset construction with rationale is possible without an external dataset even in a relatively limited model size environment (<10B).

**Strengths:**

This work shows that even if an external preference dataset does not exist, it is possible to build a preference dataset for iterative update through a relatively small-sized model with rationale and appropriate post-filtering. In particular, major voting on scores is possible even for instructions where it is difficult to determine an exact match for an answer, so it is possible to learn LLM-as-a-Judge for instructions in various domains.

**Weaknesses:**

As a preliminary study, [1] already reported that LLM-as-a-Judge can be trained iteratively without the seeds, but it is better to pre-filter the model-generated judgments that do not match human decisions for training efficiency. This suggests that complete AI-built preference datasets may not initially contain meaningful preference signals compared to filtered datasets using a small number of human preference decisions. However, this work does not include an analysis of whether oversampling of the rationale trajectory is efficient enough to bypass filtering with this external human preference. Without such justification, this work can be evaluated as lacking analysis for further research from [1].

[1] Self-Rewarding Language Models (Yuan et al., 2024)

**Questions:**

The experimental results in Table 4 show that curating the preference dataset through pairs with large margins has meaningful impacts on downstream performance, suggesting that the relative betterness in pairs is more important than the individual scores of the responses. Also, studies using pointwise assessment, such as [1], report that as training progressed iteratively, most responses were “capped” at the maximum possible scores, making it difficult to make significant pairwise differences anymore. In contrast, self-improving approaches, which use pairwise assessment, such as [2], may be relatively better on this problem, but there are no discussions about these studies. Was there any experimental evidence of choosing pointwise assessment towards pairwise assessment in this work?

[1] Meta-Rewarding Language Models: Self-Improving Alignment with LLM-as-a-Meta-Judge (Wu et al., 2024)
[2] Aligning Large Language Models by On-Policy Self-Judgment (Lee et al., 2024)

---

> ### Author Response · Authors · 2024-11-21
> **Rebuttal by authors to Reviewer Dq8w**
>
> We thank the reviewer for their detailed questions and comments. We provide our answers point-wise and seek guidance/advice on making suggested changes in the revised version.  We have uploaded a version where new changes are marked with blue text color.
> > Regarding Self-Rewarding LMs
>
> Following are the differences between Self-rewarding LMs and our propose method SRE:
> - As mentioned in Table 1, Self-rewarding LMs requires additional training data for Instruction Fine-Tuning as well as Evaluation Fine-tuning, thus being much more data costly than SRE.
> - Self-rewarding LMs do not have the ability to evaluate on the basis of a custom as well as a fine-grained likert-scale criteria
> - As mentioned by the reviewer, they have not demonstrated improvements through their recipe on a small parameter model
>
> > Regarding filtering model judgements
>
> Our proposed recipe for preference curation uses correct-answer preference pairing which is essentially using human ground truth scores to curate the preference data( see section 3.3 for more details). This ensures as pointed out by the reviewer that preference signal is not lost in the AI-generated data, as it is aligned by human-preferences.
>
> > Regarding decision of choosing pointwise assessment over pairwise assessment
>
> Thanks for the feedback. The reasons behind choosing pointiwse assessment over pairwise assessment are mainly the following:
> 1.  Pointwise assessment is inherently more challenging than pairwise assessment due to its fine-grained nature. In pointwise evaluation, judgments are made on a 5-point Likert scale, requiring precise differentiation across multiple levels. In contrast, pairwise evaluation involves selecting between only two options, which simplifies the decision-making process by reducing the complexity of the evaluation space.
> 2. Pointwise assessment represents a more general form of evaluation. A pointwise judge can naturally be used to generate pairwise preference data by comparing scores (e.g., the candidate with the higher score is chosen, while the other is rejected). Conversely, a pairwise judge is limited to binary comparisons and cannot perform the fine-grained evaluations required for a Likert-scale pointwise assessment. This highlights the broader applicability and versatility of pointwise assessment compared to pairwise evaluation.
>
> Additionally we took inspiration from the reviwer's question and have now also revised the paper to include pairwise benchmarks. We have incorporated the corresponding results in Table 3 of the revised manuscript. These results demonstrate that J_SRE exhibits robust performance across both pairwise and pointwise benchmark evaluations, along with the added benefits of being a small sized LM(<10B) and having the ability to judge based on a custom scoring criteria.

---

> ### Author Response · Authors · 2024-11-26
> **Gentle reminder for Reviewer Dq8w**
>
> Dear Reviewer,
>
> We thank you again for taking out the time to give detailed and constructive reviews. This is a gentle reminder that the discussion period is ending today. We wanted to reach out and ask if there are any more questions or suggestions regarding the rebuttals. We have uploaded a revised version of the paper with the changes marked in blue text color and have additionally incorporated pairwise evaluation benchmarks in addition to pointwise (likert scale) benchmarks in Table 3.
>
> If you find that our updates and rebuttal address your concerns, we would be grateful if you could consider revising your evaluation. We’re happy to provide further clarification or address any remaining questions you might have.
>
> Thanks again!

---

### Official Review · Reviewer_HX9K · 2024-11-03

**Soundness:** 3
**Presentation:** 3
**Contribution:** 2
**Rating:** 6
**Confidence:** 5

**Summary:**

The paper introduces Self-Rationalization as an iterative training process aimed at improving the evaluation capabilities of large language models (LLMs) when used as judges. Self-Rationalization trains the LLM by generating multiple rationales for the same input and creating preference pairs, with one judgment selected as superior. These pairs are then used for DPO fine-tuning, helping the model learn from its own rationale and score generation. They show that this method leads to improvement in rationale quality and scoring accuracy without additional human feedback, as demonstrated across multiple benchmarks.

**Strengths:**

1. The paper is written well and the experiments are comprehensive for the scope of this submission.
2. It introduces a practical approach for model alignment using self-generated rationales and DPO, minimizing reliance on human-labeled data. Unlike similar methods like Self-Taught Evaluators, it applies DPO to paired rationales, which is a sensible choice for optimizing preference alignment.

**Weaknesses:**

1. Although the model generates multiple rationales per input, there is no exploration of how the diversity of these rationales affects model performance. Ablation studies showing the impact of rationale diversity (e.g., varying the number or quality of rationales) on alignment and scoring accuracy would offer valuable insights into the robustness of the approach.
2. Rather than relying solely on high-temperature sampling to produce rejected rationales, using prompts to vary rationale quality and accuracy could generate a broader range of challenging responses, enhancing the training data.
3. Exploring different model families and parameter sizes could improve the generalization and robustness of the results, showing how well the approach scales across varying model architectures and capacities.

**Questions:**

Please refer to weaknesses

---

> ### Author Response · Authors · 2024-11-21
> **Rebuttal by authors to Reviewer HX9K**
>
> > Regarding effectiveness of rationales and rationale diversity
>
> To assess the effectiveness of rationales, we conduct additional ablation experiments to demonstrate that rationales play a critical role in both alignment and reasoning within our proposed framework. Specifically, we examine the impact of rationale quality on preference data curation. In this experiment, we prompt  $J_\text{SFT}$ to generate low-quality rationales, while ensuring that the chosen and rejected items still match the ground truth scores.  In this setup, we keep the chosen rationale of poor quality and the rejected rationale from $J_\text{SFT}$, but the score alignment with the ground truth remains intact. This design allows us to isolate and highlight the importance of rationale quality, over mere score correctness, in the alignment process (i.e., DPO). We then compare the performance of this modified model after DPO with the original  $J_\text{SFT}$ model and the results in Table 4 show the degradation of performance demonstrating that significance of generated rationales(more than scores) in the alignment(DPO):
> | Model                  | Biggen Human Corr | Biggen GPT-4 Corr | Reward Bench Chat  | Reward Bench Chat Hard | Reward Bench Safety | Reward Bench Reasoning | Weighted Average |
> |------------------------|--------------|--------------|--------------------|-------------------------|---------------------|-------------------------|-------------------|
> | **J_SFT**              | 0.49         | 0.60         | 0.79               | 0.53                    | 0.83                | 0.66                    | 0.69              |
> | **Modified J_SRE (preference reversal, margin 0)** | 0.36        | 0.41         | 0.84             | 0.49                  | 0.81               | 0.60                   | 0.65           |
>
> > Generalization of recipe to different models:
>
> Thanks for the suggestion! We have reproduced our recipe for the Qwen-2 model and the details are in A.4. The results as shown below shows that our proposed recipe improves over SFT and off-the-shelf models:
> | Benchmark      | Off-the-shelf | SFT   | DPO   |
> |----------------|---------------|-------|-------|
> | BigGen bench   | 0.519         | 0.603 | 0.642 |
> | RewardBench    | 0.646         | 0.660 | 0.680 |

---

> ### Author Response · Authors · 2024-11-26
> **Gentle reminder to Reviewer HX9K**
>
> Dear Reviewer,
>
> We thank you again for taking out the time to give detailed and constructive reviews. This is a gentle reminder that the discussion period is ending today. We wanted to reach out and ask if there are any more questions or suggestions regarding the rebuttals. We have uploaded a revised version of the paper with the changes marked in blue text color. In particular we have incorporated additional experiments regarding rationale diversity in Table 4 and have reproduced the proposed recipe with results shown in A.4 in the updated paper.
>
> If you find that our updates and rebuttal address your concerns, we would be grateful if you could consider revising your evaluation. We’re happy to provide further clarification or address any remaining questions you might have.
>
> Thanks again!

---

### Official Review · Reviewer_QcQc · 2024-11-04

**Soundness:** 3
**Presentation:** 2
**Contribution:** 3
**Rating:** 6
**Confidence:** 4

**Summary:**

The goal of this paper is to enhance the performance of LLM-as-a-judge, ensuring that high-quality responses receive higher scores and low-quality responses receive lower scores.

The method is the first to improve the performance of the judge model by RLAIF: an iterative training process comprising SFT and DPO on self-curated preference data.
Specifically, the model is prompted to output rationales when evaluating each content, and multiple samples are taken for the same question. The preferences among different rationales are determined based on the whether the model’s judgments and the ground truth are aligned. This preference data is then used for two rounds of training, with the updated model regenerating the preference dataset after each round.

Experiments demonstrate a slight improvement in correlation with human or GPT4 scoring after applying this SFT+DPO training.

**Strengths:**

Originality: The paper introduces a new pipeline consisting of data curation and alignment training to enhance LLM's ability as a judge. It builds on existing judge models like Promethus and goes further by implementing RLAIF fine-tuning.

Quality: The paper considers multiple benchmarks including Reward Bench, BiGGen Bench, and Feedback Bench, and shows good performance against both comparable and larger models.

Significance: overall, the paper has a potential impact for practitioners on the field of LLM-based evaluation as it implements the pipeline of RLAIF and shows its performance. This is particularly important for applications where understanding and alignment with human reasoning are critical. Furthermore, it reduces the reliance on extensive human-labeled data, making it a scalable alternative to traditional reinforcement learning from human feedback.

**Weaknesses:**

Clarity: the paper's presentation can be further improved by including more details. Please see the questions below.

Originality: The training dataset and the SFT training method is proposed by Promethus paper, and the pairwise/pointwise scoring functions and fine-grained criteria are also considered in this paper. The contribution of this paper related to method novelty lies on the heuristic used for preference selection.

**Questions:**

Writing:

Q1: The paragraph starting at line 52 provides a vague explanation for the motivation, and the logical flow between sentences feels disconnected. Why does producing and reflecting on rationales enhance the LLM’s scoring ability? How do iterative generations and improvements of rationales lead to better alignment and calibrated evaluations? I suggest adding references or breaking down the logical chain in more detail to improve clarity.

Q2: The related works section is unclear in places. For example, in the caption for Table 1, what does “whether synthetic seed prompts need to be generated” mean? Does it refer to not needing an extra training dataset?

Q3: Which specific preference selection method is used in Tables 2 and 3? Is it the Correct-Answer Preference Pairing method? Additionally, starting from line 371, how is the margin defined? Is it determined by taking the highest scoring sample and selecting those with scores within a certain margin? If all samples’ score difference is below this margin, how are they handled? Are they discarded?

Q4: There are several incorrect uses of citations; for instance, in line 483, parentheses should not be used.

Q5: In the related works section, why does the subsection on LLM-as-a-judge and reward models emphasize DPO? Are there no methods that utilize RLHF?

Experiments:

Q6: How is the score distribution for the majority vote calculated in line 213?

Q7: What does the statement “use ground truth score to guide the selection” in line 293 mean?

Q8: In line 311, the statement “require fewer training samples and compute resources”—what models is this being compared to? Could more explanation be provided to support this claim?

Q9: What experimental evidence supports the claim in line 317 that long rationales can lead to external noise and complexity in token predictions?

Q10: Is the statement in line 322 reversed?

Q11: How much data was used for evaluation in Figure 2?

Q12: What specifically does “ground truth rationale” refer to in line 345?

Minor:

Q13: The citation for Kim 2023 and Kim 2024a in line 247 seems to reference the same work.

---

> ### Author Response · Authors · 2024-11-20
> **Rebuttal by authors to Reviewer QcQc**
>
> We thank the reviewer for his very detailed comments and suggestions. We provide our answers question-wise and seek guidance/advice on making suggested changes in the revised version. We have uploaded a version where new changes are marked with blue text color.
>
> **Q1**: Thanks for the suggestion. We have improved the writing for clarification.
>
> **Q2**: Thanks for clarifying the statement. We have revised the caption in Table 1 to make it clear that no additional external training data is required for our proposed recipe.
>
> **Q3**: Thanks for pointing this out. We have now explicitly stated the preference curation method we have used. We use the correct-answer preference pairing margin>=2 preference curation method across all the experiments as it performed the best in the ablation experiments for different preference selection methods as shown in Table 5.
>
> **Q4**: We have revised the paper to correct this.
>
> **Q5**: Thanks for the feedback. We have highlighted the fact that performing DPO with rationales for LLM-as-Judge training is under-explored in the research community. There are methods that use RLHF, but as RLHF is very expensive as compared to RLAIF, efficient methods need to be proposed for Judge training as well as other downstream tasks.
>
> **Q6**: We generate N judgements for the same input data point i.e. run inference from the given Judge model N times to obtain the judgments(consisting of rationale+score). The obtained score distribution is then used to curate the preference data.
>
> **Q7**: The line  “use ground truth score to guide the selection” means that in the  preference data curation, from all the possible pairs created by Self-consistency/majority-voting, chosen judgment is kept as the one which has score matching the ground truth score and rejected judgment as the one which does not match ground truth score.
>
> **Q8**:  We have revised the given line in the paper to highlight what we are comparing against. As compared to SFT, DPO is more compute and data efficient as:
>  - DPO uses preference pairs (e.g., good vs. bad responses), requiring less data and consequently compute compared to SFT, which needs large, manually labeled datasets
> - While training for other post-SFT methods like self-consistency, best-of-N method, in order to come close to the performance of their counterpart DPO models, we observed that it took almost 2.5x more time and compute and 6x more data than DPO.
>
>
> **Q9**: As demonstrated by Chen et al. [1], incorporating rationales during supervised fine-tuning (SFT) can dilute the training signal. This occurs because the model may focus on irrelevant details and learn spurious correlations, an issue that does not arise with DPO, as shown in Table 4. Notably, SFT_without_rationale outperforms SFT_with_rationale. In contrast, DPO_without_rationale (SRE_wo_rationale) underperforms compared to DPO_with_rationale (SRE_with_rationale).
>
> **Q10**:Thank you for pointing that out! We have revised the given line for clarity. It now explicitly states that when the model is prompted to provide only a score without accompanying rationale, J_SRE (prompted  to output only a score and not rationale) performs worse than J_SFT in this specific scenario, showing that rationales are more significant for DPO than SFT.
>
> **Q11**: The dataset size was 1000 in total. 500 samples each was randomly sampled from BigGen Bench and Feedback Bench, and given to three annotators independently.
>
> **Q12**: Thanks for pointing it out. We have revised the given line for increased clarity.
>
> **Q13**: The works referenced are from the same author(s), but are different. Biggen Bench was created and improved over Prometheus-family models.
>
> [1] Zhipeng Chen, Kun Zhou, Wayne Xin Zhao, Junchen Wan, Fuzheng Zhang, Di Zhang, and Ji-Rong Wen. Improving large language models via fine-grained reinforcement learning with minimum editing constraint

---

> ### Author Response · Authors · 2024-11-26
> **Gentle reminder to Reviewer QcQc**
>
> Dear Reviewer,
>
> We thank you again for taking out the time to give detailed and constructive reviews. This is a gentle reminder that the discussion period is ending today. We wanted to reach out and ask if there are any more questions or suggestions regarding the rebuttals. We have uploaded a revised version of the paper with the changes marked in blue text color.
>
> If you find that our updates and rebuttal address your concerns, we would be grateful if you could consider revising your evaluation. We’re happy to provide further clarification or address any remaining questions you might have.
>
> Thanks again!

---

### Official Review · Reviewer_V1QD · 2024-11-04

**Soundness:** 3
**Presentation:** 2
**Contribution:** 2
**Rating:** 5
**Confidence:** 4

**Summary:**

The paper introduces Self-Rationalization, a method to improve the performance of Large Language Models (LLMs) used as judges by enabling them to iteratively train on their own rationales. Through a process involving Direct Preference Optimization (DPO), the model generates multiple judgments for the same input, selects preferred rationales, and fine-tunes itself without additional human-labeled data. Experimental results show that Self-Rationalization outperforms existing baselines on several benchmarks, indicating that this targeted training enhances the model's judgment capabilities.

**Strengths:**

- This paper introduces a practical framework for enhancing the judgment capabilities of large language models (LLMs) through Self-Rationalization, an iterative self-training process.
- The approach is both practical and resource-efficient, requiring no additional human-labeled data, making it broadly applicable to LLM-based evaluation tasks.

**Weaknesses:**

- The baseline comparisons are insufficient, as the authors should evaluate the proposed method against more iterative self-improvement approaches (e.g., Self-Taught Evaluators) to provide a broader perspective on its effectiveness.
- The paper does not convincingly demonstrate the impact of rationales on enhancing judgment capabilities. As shown in Table 3, the performance gain from including rationales is modest (an increase from 72.0 to 74.0 on RewardBench). Additionally, when compared to the pairwise-trained Skywork-Critic-Llama-3.1-8B (which achieves 89.0 on RewardBench), the use of rationales does not offer a clear advantage.

**Questions:**

1. As described in line 258, the temperature is set to 1.0 when sampling predictions. Does this setting ensure a strong correlation between the generated rationales and scores? For instance, what is the distribution of scores when multiple scores are generated based on the same rationale? Could this affect the quality of preference pairs, where a high-quality rationale might receive an inaccurate score due to the high temperature, leading it to be incorrectly optimized as a negative example?
2. In line 252, why was weight-merging chosen over training on a mixed dataset?
3. In line 259, what is the reason for using different data quantities across the two iterations?
4. What impact would additional iterations have on the model's performance?

---

> ### Author Response · Authors · 2024-11-20
> **Rebuttal by authors to Reviewer V1QD**
>
> We sincerely thank you for spending the time to review our work and for providing constructive reviews. We provide our answers point-wise and seek guidance/advice on making suggested changes in the revised version.  We have uploaded a version where new changes are marked with blue text color.
> > Baseline comparisons
>
> Thanks for the feedback. In Table 1, we highlight the differences of other self-improvement approaches like Self-Taught evaluators, Self-rewarding LMs etc. Specifically, Self-taught evaluators do not have the ability to take in custom fine-grained evaluation criteria, and additionally are of much bigger parameter size(i.e. 70B) as compared to $J_\text{SRE}$ which is 8B. Furthermore, the evaluation setting used by Self-taught evaluators is a pairwise setting in which the model is prompted to choose the better response between 2 candidate responses, unlike $J_\text{SRE}$ which is evaluated in the more stricter setting of pointwise likert-scale evaluation(more details of settings in Background section). Similarly, we do not compare our results with Skywork-Critic-Llama-3.1-8B, as it is a pairwise model, and cannot perform fine-grained evaluation with custom scoring criteria.
>
> > Effect of temperature in preference data curation
>
> Thank you for pointing this out. In Table 5, we demonstrate that the meta judge—designed specifically to assess the quality of rationales without focusing on scores and using a high-temperature setting—achieves performance comparable to the J_SRE preference curation method. This finding highlights that maintaining a high temperature does not result in cases where judgments with incorrect scores but high-quality rationales are incorrectly categorized as negative examples.
>
> > Weight merging vs Dataset mixtures
>
> Prometheus 2[1] have demonstrated experimentally the effect of single-format training, joint-training(i.e. Merging the datasets) and weight merging. Similar to their results we observed that linear merging the pairwise and the pointwise models improved the task performance of both pointwise and pairwise tasks more than mixing the dataset.
>
> > DPO sample sizes across iterations
>
>  Empirically, we observed that different datasets required progressively less data to fine-tune at each stage, resembling a curriculum learning approach. As the model improves with each iteration, its performance becomes more refined, leading to more accurate judgments. Consequently, it becomes more challenging to identify rejected data points in the preference pairs where the model's score is incorrect. As the model's accuracy increases, the number of such 'incorrect' data points decreases, reducing the total amount of preference data required.
>
> > Impact of more iterations in Iterative DPO
>
> Thanks for pointing that out, we have now clarified the reason in the paper.  Our experiments showed diminishing returns with additional training iterations, as the model's accuracy plateaued and minimal improvements were observed.
>
>
> [1] Prometheus 2: An Open Source Language Model Specialized in Evaluating Other Language Models Kim et. al.

---

> ### Author Response · Authors · 2024-11-26
> **Gentle reminder to Reviewer V1QD**
>
> Dear Reviewer,
>
> We thank you again for taking out the time to give detailed and constructive reviews. This is a gentle reminder that the discussion period is ending today. We wanted to reach out and ask if there are any more questions or suggestions regarding the rebuttals. We have uploaded a revised version of the paper with the changes marked in blue text color.
>
> If you find that our updates and rebuttal address your concerns, we would be grateful if you could consider revising your evaluation. We’re happy to provide further clarification or address any remaining questions you might have.
>
> Thanks again!

---

### Official Review · Reviewer_o4Kb · 2024-11-04

**Soundness:** 2
**Presentation:** 2
**Contribution:** 2
**Rating:** 5
**Confidence:** 4

**Summary:**

This paper presents Self-Rationalization, a method that enhances the rationale generation capabilities of LLM judges, thus improving the RLAIF process. Aligned with the standard RLAIF framework, this approach samples multiple potential scores with associated rationales, curates data from these samples, and applies DPO for LLM refinement. By iteratively refining (e.g., through two iterations), this framework shows marked improvements over baseline LLMs and prior RLAIF methods.

**Strengths:**

- Self-refinement of LLMs through RLAIF is a significant research topic.
- Generating rationales for LLM judges enhances both performance and interpretability.
- The proposed approach refines and improves baseline LLMs.

**Weaknesses:**

**Limited technical novelty**

The concept of generating rationales for LLM judges (or reward models) has been explored in recent works [1,2]. Both papers were uploaded to arXiv in August 2024, making them concurrent studies; therefore, it is okay not to include experimental comparisons with them. However, given these existing works, it is essential to identify the unique contributions of this paper. Including a discussion on the pros and cons of different approaches would be valuable.

[1] Ankner et al. Critique-out-Loud Reward Models. arXiv 2024.\
[2] Zhang et al. Generative Verifiers: Reward Modeling as Next-Token Prediction. arXiv 2024.

---
**Limited improvement in performance**

The performance gains are not significant, despite the additional computation involved in two iterative refinements, which include complex procedures for sampling scores and rationales. For example, in Fig. 2, the win rate of the proposed SRE over SFT is only 55%, a modest 5% improvement over a random guess. Similarly, the results in the tables appear limited. In Table 2, the improvement over single-stage DPO is minimal, raising questions about the effectiveness of the generated rationales. Additionally, in line 322, where it states, “We observe that $J_{SRE}$ performs significantly worse than $J_{SFT}$,” it should instead read significantly better, not worse?

---
**Writing could be further polished**

For instance, the first paragraph of the introduction introduces the concept of RLAIF, but the second paragraph revisits the limitations of RLHF, which feels redundant. These two paragraphs could be reorganized to convey distinct points. Other sections could also be reorganized. For example, in the Background section, it’s unclear why pointwise or pairwise assessments are emphasized as being so important.

**Questions:**

N/A

---

> ### Author Response · Authors · 2024-11-20
> **Rebuttal by authors to reviewer o4Kb**
>
> We  thank you for spending the time to review our work and for providing constructive reviews. We provide our answers point-wise and seek guidance/advice on making suggested changes in the revised version. Additionally, we have uploaded a version where new changes are marked with blue text color.
> > Comparison to existing works
>
> We appreciate your efforts in bringing the relevant papers to our attention. We have added a discussion highlighting the pros and cons of the given approach in the Related work section (Rationales subsection) and have marked the added text in blue.
> > Regarding limited performance gains and effectiveness of rationales
>
>  In the win-rate comparison between $J_\text{SFT}$ and $J_\text{SRE}$, we adopted a stricter evaluation approach. Specifically, ties were included in the win-rate comparison plots presented in Figure 2 and resolved through random selection. When ties are excluded, $J_\text{SRE}$ achieves a win rate of 61% against $J_\text{SFT}$. To assess the effectiveness of rationales, we conduct additional ablation experiments to demonstrate that rationales play a critical role in both alignment and reasoning within our proposed framework. Specifically, we examine the impact of rationale quality on preference data curation. In this experiment, we prompt  $J_\text{SFT}$ to generate low-quality rationales, while ensuring that the chosen and rejected items still match the ground truth scores.  In this setup, we keep the chosen rationale of poor quality and the rejected rationale from $J_\text{SFT}$, but the score alignment with the ground truth remains intact. This design allows us to isolate and highlight the importance of rationale quality, over mere score correctness, in the alignment process (i.e., DPO). We then compare the performance of this modified model after DPO with the original  $J_\text{SFT}$ model and the results in Table 4 show the degradation of performance demonstrating that significance of generated rationales(more than scores) in the alignment(DPO):
> | Model                  | Biggen Human Corr | Biggen GPT-4 Corr | Reward Bench Chat | Reward Bench Chat Hard | Reward Bench Safety | Reward Bench Reasoning | Weighted Average |
> |------------------------|--------------|--------------|--------------------|-------------------------|---------------------|-------------------------|-------------------|
> | **J_SFT**              | 0.49         | 0.60         | 0.79               | 0.53                    | 0.83                | 0.66                    | 0.69              |
> | **Modified J_SRE (preference reversal, margin 0)** | 0.36        | 0.41         | 0.84             | 0.49                  | 0.81               | 0.60                   | 0.65           |
>
> > Additionally, in line 322, where it states, “We observe that  $J_\text{SRE}$ performs significantly worse than $J_\text{SFT}$,” it should instead read significantly better, not worse?
>
> Thank you for pointing that out! We have revised the given line for clarity. It now explicitly states that when the model is prompted to provide only a score without accompanying rationale, $J_\text{SRE}$ (prompted  to output only a score and not rationale) performs worse than $J_\text{SFT}$ in this specific scenario, showing that rationales are more significant for DPO than SFT.
>
> > Polishing writing
>
>  Thanks for the suggestion. We have improved the writing to give more context and have reorganized the flow to improve readability and understanding.

---

> ### Author Response · Authors · 2024-11-26
> **Gentle reminder to Reviewer o4Kb**
>
> Dear Reviewer,
>
> We thank you again for taking out the time to give detailed and constructive reviews. This is a gentle reminder that the discussion period is ending today. We wanted to reach out and ask if there are any more questions or suggestions regarding the rebuttals. We have uploaded a revised version of the paper with the changes marked in blue text color. In particular we have incorporated additional experiments regarding effectiveness of rationale in Table 4 and have also added more comparisons to Related work as well as polished the writing.
>
> If you find that our updates and rebuttal address your concerns, we would be grateful if you could consider revising your evaluation. We’re happy to provide further clarification or address any remaining questions you might have.
>
> Thanks again!

---

> > ### Comment · Reviewer_o4Kb · 2024-12-01
> > **Response to the Rebuttal**
> >
> > Thank you for the rebuttal. However, the performance gains are still not significant. Excluding ties from Figure 2 is nonsense, as it implies $J_\text{SFT}$ and $J_\text{SRE}$ are on par in many cases. Additionally, the ablation study is not convincing enough, as the final effect combining rationale quality and score correctness is the real matter in practice. Based on this, I maintain my original rating of borderline rejection.

---

### Meta-Review · Area_Chair_gpdY · 2024-12-18

**Metareview:**

This paper proposes Self-Rationalization, a novel framework to improve LLM judges by generating and refining rationales through the RLAIF. The approach attempts to strengthen the model’s evaluation capabilities without relying on additional human-labeled instruction-tuning data. While the concept of producing rationales for judges aligns with recent trends in LLM-based evaluation, the technical contributions appear not novel enough, and the improvements over baseline methods are incremental. Also, the methodology borrows from the existing self-rewarding approach, adapting the training strategy for a smaller-scale model rather than demonstrating clear advances over that baseline. The potential benefits of rationales remain unclear, as performance gains are slight and the rationale-enhanced model does not strongly outperform simpler approaches.
The paper does not extensively compare against other iterative self-improvement techniques, raising questions about its relative effectiveness. The writing and organizational structure could also be clarified. Overall, the limited performance gains and uncertain advantages relative to existing methods reduce the paper’s potential impact.

**Additional Comments On Reviewer Discussion:**

Low-quality reviews are considered carefully in the decision process, and I did not highly consider feedback from reviewers who did not engage during this phase. During the rebuttal period, the authors emphasized their resource-efficient strategy, training two models (pointwise and pairwise) without extra instruction-tuning data, and highlighted improvements over baseline LLMs. However, reviewers noted that the approach parallels self rewarding LLMs and that it is not evident how their method would compare if the baseline were similarly scaled down or tested under equivalent conditions. Although the authors clarified certain methodological choices, such as weight merging and sampling parameters, the reviewers remained unconvinced about the significance of these refinements. The questions about quality and insufficient comparisons to other prior iterative approaches, led to skepticism that the proposed technique delivers meaningful breakthroughs.

---

### Decision · Program_Chairs · 2025-01-22

Reject